# GenPad: A Highly Efficient Roadmap for the Development of a New Rapid, Highly Sensitive, and Portable Point-of-Care Testing System for Nucleic Acid Diagnostics in Japan

**DOI:** 10.3390/diagnostics15162020

**Published:** 2025-08-12

**Authors:** Oleg Gusev

**Affiliations:** Intractable Disease Research Center, Graduate School of Medicine, Juntendo University, 2-1-1 Hongo, Bunkyo-ku, Tokyo 113-8421, Japan; o.gusev.fo@juntendo.ac.jp

**Keywords:** POCT, COVID-19, nucleic acid testing, isothermal amplification, RNA

## Abstract

From the corona virus pandemic in Japan that started with the “Diamond Princess” accident, it became clear that rapid detection, a high sensitivity, multiple diagnostic items, one-step one-base point mutation detection, a fast speed of system development, portability (small size and light weight), full automation, random access, and other conditions are required for future point-of-care testing systems. The Eprimer-SmartAmp technology that was developed possesses characteristics fully aligned with these requirements. Building upon this platform, the “GenPad” system was subsequently established. The GenPad system is widely applicable not only to emerging foreign infectious diseases, but also to cancer, lifestyle-related diseases, and other areas of healthcare through telemedicine and intraoperative nucleic acid diagnoses. In collaboration with telecommunication systems, GenPad is expected to contribute to the establishment of a smart medical city with a countermeasure against emerging foreign infectious diseases, where individuals can check their own health conditions in all healthcare areas.

## 1. Introduction

The COVID-19 pandemic underscored the critical importance of rapid, portable, and highly sensitive nucleic acid diagnostics. Traditional PCR-based tests, although accurate, often require specialized laboratories and lengthy processing times, limiting their utility in immediate decision-making contexts. Point-of-care testing (POCT) technologies emerged as essential tools to bridge this gap, enabling quick on-site diagnostics in hospitals, community settings, transportation hubs, and remote locations. One of the early scenarios highlighting the urgent need for reliable POCT was the “Diamond Princess” cruise ship incident in Japan, where rapid testing could have significantly improved patient management and containment measures. The global COVID-19 pandemic has highlighted critical gaps in the current diagnostic infrastructure, particularly the need for rapid, decentralized, and highly sensitive testing solutions to support public health responses.

In this opinion article, I discuss the scientific basis, practical features, and potential applications of the GenPad POCT system, recently developed based on the Eprimer-SmartAmp isothermal amplification technology. POCT technologies are being increasingly recognized as essential components of modern healthcare systems, enabling timely diagnoses and immediate clinical decision-making without reliance on centralized laboratories [1]. I provide a concise overview of the underlying technology, evaluate its practical advantages and current limitations, compare GenPad to existing nucleic acid POCT platforms, and explore its prospective roles within telemedicine frameworks and public health infrastructures. This article aims to stimulate further scientific discussion on optimizing nucleic acid diagnostics and their integration into future healthcare ecosystems.

## 2. The Fight Against the New Coronavirus Started with the “Diamond Princess” Incident

In January 2020, the coronavirus pandemic in Japan began with the arrival of the “Diamond Princess” cruise liner at Yokohama Port. Due to an outbreak of a new coronavirus onboard the ship, quarantine authorities did not allow passengers to land on the ship for two weeks after it entered Yokohama Port, and approximately 3500 people were confined onboard [2]. During this time, the new coronavirus continued to spread throughout the ship, as it was not possible to identify who was infected. At that time, our group at RIKEN received a letter from the governor of Kanagawa Prefecture, Kuroiwa. After receiving a strong request to make a new coronavirus test kit, we immediately began developing new coronavirus test reagents. It took approximately two weeks to create a kit that could be used for corona detection in the laboratory, but it was not possible to develop reagents for use onboard the ship.

This experience yielded several critical insights. In particular, we understood the conditions that a testing system of a Smart Medical City should meet in order to handle emerging outpatient infectious diseases, and we also realized that our current testing system at that time did not meet those conditions. This article summarizes the lessons learned from facing the coronavirus pandemic that started with the Diamond Princess.

## 3. System for Dealing with Emerging Infectious Diseases

Initially, when test reagents were developed, the developers considered the performance criteria that the reagents should meet. The reagents must meet the following conditions: (1) rapidity, (2) a high sensitivity, (3) multiple test items, (4) one-step (single-base) mutation detection, and (5) a rapid system development speed. In addition, the platform that detects it must be (1) portable (small and lightweight), (2) fully automatic (just turn it on), and (3) randomly accessed (start testing as soon as a sample is obtained) (Figure 1).

The conventional system of sending a test to a testing center and waiting for the results does not allow for on-the-spot testing, so rapid testing, or point-of-care testing, is essential. The necessity of this condition can be easily imagined when we assume an inspection on board the Diamond Princess. In particular, there are many cases where there is not enough time to transport the specimen to a diagnostic lab. Therefore, the system itself must be portable. In addition, in order to be able to use it on the spot, it must be easy to operate [3,4,5,6].

One of the characteristics of the new coronavirus infection is that, not only it is highly contagious, but many infected people are also asymptomatic, and they are often infected before they develop symptoms. This point is one of the reasons why the infection has spread all over the world [7]. Selecting presymptomatic infected individuals (who are shedding germs) from close contacts was extremely important for preventing clusters. In particular, those who shed the virus before the onset of symptoms have a low copy number of the virus, so a highly sensitive test is required.

An issue in the experience of the coronavirus pandemic is the speed at which viruses mutate. Viruses are suddenly becoming more infectious, vaccines are no longer effective, and viruses are becoming resistant to antiviral drugs [8]. Global experience shows that the rate of mutation of the new coronavirus is high [9], and the tests and vaccines we develop need to catch up with the mutations of the virus. In the medical field, there are many seriously injured patients, so it is necessary to immediately detect drug resistance by POCT testing. To this end, there is a strong need in medical practice to detect mutations using a one-step procedure.

During the coronavirus pandemic, there was a need for a system that would enable random access, allowing testing to begin immediately in the order in which people access it, at meetings where people gather together, and on transportation vehicles such as airplanes. A test system that satisfies all of these conditions is a planform that can respond to emerging outpatient infectious diseases [10].

## 4. Development of Eprimer-SmartAmp Technology—The Basic Chemical Reaction for the GenPad Platform

We have previously described SmartAmp [11] as a single-nucleotide mutation-specific rapid nucleic acid amplification method. This method is an isothermal polymerase chain reaction (isothermal PCR) that specifically amplifies any mutation in any part of a nucleic acid [12].

The reagents for the Eprimer-SmartAmp method achieved rapidity, a high sensitivity, one-step (one-base) mutation detection, and a rapid system development speed, among the above conditions. Furthermore, because it is an isothermal PCR method, it does not require normal temperature cycling and only needs to be heated with a heater, and because it consumes little electricity, it can be powered by a mobile phone battery, making it more compact and portable [13].

### 4.1. Principle and Features of SmartAmp as a Sensor for Nucleic Acids

We have been developing SmartAmp since around 2002. A feature of SmartAmp’s primers is that they use an asymmetric primer pair of a turn-back primer and a folding primer to achieve single-base-specific nucleic acid amplification. By combining the methods of fluorescently labeling the target nucleic acid, the signal became extremely strong and the S/N ratio improved (Figure 2). The approach and sensitivity of SmartAmp has already been acknowledged as one of the most sensitive for point mutations [14,15,16].

The turn-back primer creates a loop and the next turn-back primer hybridizes to that loop, priming the polymerase and the 3′ ends of both the turn-back and folding primers and the hairpin loop of the folding primer. A mechanism in which polymerase primes from the 3′ end produces long molecules. Single-base mutation-dependent amplification (one-step (single-base) mutation detection) was achieved quickly, with a high sensitivity and under isothermal conditions. As shown, 600 copies of the target nucleic acid were produced approximately 17 min after the actual reaction started. The mutation could be typed the moment amplification was detected [13]. It is worth mentioning that the reaction itself amplifies nucleic acids with a sensitivity comparable to that of temperature-cycling PCR, so it can be used as a basic reaction for highly sensitive and rapid nucleic acid detection methods [17,18,19,20,21].

The exciton primer (Eprimer) attaches a linker with two arms to the fifth carbon atom of the pyrimidine of the oligonucleotide that serves as primer. It has a structure in which two fluorescent dye molecules are covalently bonded (Figure 3). In this structure, when the oligonucleotide is not hybridized with the target sequence, the π electron orbitals of the two molecules are close and quench. Additionally, when thiazole orange hybridizes with the target sequence, it has the property of fitting into the major groove of the double helix. As the two molecules are stuck together, they separate and emit fluorescence. When hybridized or not, there is a switch between quenching and emitting light, resulting in an extremely high signal-to-noise ratio and a strong signal. Therefore, the signal after amplification using the SmartAmp method can be detected visually. In addition, it is possible to perform multi-color measurements with up to four colors and quantitative measurements using the Ct value of amplification [22,23].

By using mutation-specific primers, mutations can be determined when the amplification determination threshold is exceeded. Since the amplified signal itself becomes the mutation determination signal, there is no need to wait for PCR amplification to reach a plateau before subjecting it to the mutation determination reaction, as in the conventional method.

#### Principle of SmartAmp Amplification and the Turn-Back Primer

SmartAmp amplifies DNA at a single, constant temperature (≈63–65 °C) by using a strand-displacing polymerase, so it dispenses with the denaturation/annealing cycles required in conventional PCR and delivers results in minutes [24].

Its specificity comes from the turn-back primer (TB primer): after extension, the 5′ tail of the primer is complementary to a downstream segment of the nascent strand and folds back to create a hairpin loop. This loop becomes a fresh priming site and drives rapid, self-propagating amplification. A single-base mismatch at the TB primer’s 3′ end prevents loop formation, so amplification stops—allowing for one-step SNP or variant calls [25].

Additional primers sustain and report the reaction: a folding primer extends the loop, a boost primer accelerates the early phase, and an exciton primer (Eprimer^®^) carries a quenched fluorophore pair that fluoresces only when perfectly hybridized, giving a bright, real-time read-out without extra probes [25]. SmartAmp has already proven to be practical in diverse settings, from rapid SARS-CoV-2 screening [26] to saliva-based COVID-19 diagnostics in the clinic [27].

### 4.2. Conditions Required for Sample Pretreatment

In order to fully automate temperature cycling and isothermal PCR, a technique called direct detection is generally used, in which a sample is placed in a preparation solution without purifying the nucleic acid, and a portion of it is used for the reaction. However, with this method, only a small portion of the viral nucleic acid in the sample can be added to the reaction solution. It is true that the direct method is convenient for simplicity and automation, but it completely sacrifices sensitivity, which is the most important aspect of using PCR. The specifications of the systems that use these direct-method kits state the detection sensitivity of the target nucleic acid (viral nucleic acid) per unit volume of pretreatment solution. However, this makes no sense at all, and what is required at the testing site is the sensitivity from the entire sample that can be collected from the human body, and the sensitivity per unit pretreatment liquid volume of the sample diluted with the pretreatment liquid. It makes no sense to evaluate this. Therefore, in developing GenPad, we developed a fully automatic system that can concentrate and purify nucleic acid from the entire sample injected into a cartridge and detect how much target viral nucleic acid is present in the entire sample.

An exciton primer (Eprimer^®^) has a structure in which a linker is attached to a short single-stranded DNA (oligonucleotide) that is complementary to the target nucleic acid, and a fluorescent dye (such as thiazole orange) that functions as an intercalator is attached. If the oligonucleotide is not hybridized with the target nucleic acid, the two fluorescent dye molecules interact and quench each other. Additionally, when the oligonucleotide hybridizes with the target nucleic acid, the fluorescent dye acts as an intercalator, causing the two molecules to separate and no longer interact, and the fluorescent dye begins to emit light. Quenching and emission are actively turned on and off depending on whether or not it hybridizes to the target nucleic acid, making it possible to obtain a high signal-to-noise ratio [13].

### 4.3. COVID-19 SmartAmp Reagent: Sensitivity and LOD

During the Diamond Princess incident mentioned above, a new coronavirus detection kit was created using the Eprimer-SmartAmp method. We created a reagent that has the same sensitivity as temperature-cycling PCR and reacts quickly. Even 10 copies of viral-genome RNA can be detected in 30 min [13]

During the new 2009 influenza pandemic, it took six months to develop a kit like this, but now the kit can be completed in two weeks. I realized that the technology that accelerates the speed of kit development is also extremely important [28].

A new coronavirus detection reagent, QITO, was developed using the Ejprimer-SmartAmp reagent. The primers are designed to target the Nsp15 gene in the viral genome, which has the lowest mutation rate. In all cases, 10 copies of the viral genome were detected in 30 min. Required time (reaction): SmartAmp can make a “negative” judgement in 30 min (40 min for drug product inserts). Depending on the target, a positive judgment can be made in 12 to 15 min. The detection limit (minimum number of viruses in a detection reaction tube) is 50 copies/tube (actually 20 to 25 copies), according to the LOD (drug information) of the drug package insert [13].

## 5. Development of GenPad POCT Automated Platform

### 5.1. Conditions Required for the Platform

The Diamond Princess incident served as a practical demonstration of the urgent need for diagnostic systems that can operate directly at the point of care. The experience emphasized not only the importance of rapid and sensitive testing, but also the necessity for systems that are operationally independent, require minimal user input, and can be deployed immediately upon sample collection. To achieve this, first of all, in order to realize a portable device, the device that serves as the platform must be made smaller and lighter. Conventional temperature-cycling PCR consumes power when raising and lowering the temperature. However, our device uses isothermal PCR rather than temperature-cycling PCR as its basic principle, so the temperature remains constant and there is no need to raise or lower the temperature. Therefore, the device does not consume much power and can be operated on the battery of a mobile phone. In other words, the fact that SmartAmp uses isothermal PCR has greatly contributed to making the device smaller, lighter, and more portable.

### 5.2. Characteristics of GenPad

We designed the GenPad device as a platform that satisfies the conditions described in Section 4.1 and Section 4.2 above, and created the GenPad system by combining it with the GenPad cartridge.

#### 5.2.1. Cartridge Sensitivity and Number of Reaction Slots

Table 1 shows the features of GenPad. As mentioned above, all of the viral nucleic acids present in the sample to be tested are concentrated, recovered, and subjected to the reaction, so the detection sensitivity within a single reaction tank is almost the same as that of a PCR test in a laboratory. Since one cartridge has four reaction vessels, four targets can be detected simultaneously. One reaction chamber has a detection sensitivity of about 60 to 250 copies (300 to 1000 copies/cartridge), including the efficiency of nucleic acid purification. In particular, the US CDC’s Diagnostic Test for COVID-19, 16 August 2020, which was developed when the coronavirus pandemic began, recommended a combo menu of similar diseases rather than a single detection system for the new coronavirus (simultaneously detecting multiple target viruses to be differentiated with one test). A problem has been reported in that a negative result of the coronavirus test alone cannot provide a definitive diagnosis, even when patients with a fever are seen, which does not lead to benefits for patients.

#### 5.2.2. Internal Structure of the GenPad Device

To clarify the internal components of the GenPad nucleic acid testing device, the main elements are described briefly here. GenPad consists of a disposable cartridge and a portable detection unit. The cartridge contains four independent reaction chambers pre-loaded with all the necessary reagents for SmartAmp amplification. A single sample inlet allows for the injection of biological material, after which the sample is automatically distributed to the reaction chambers. Within the detection unit, a heating module maintains the constant isothermal conditions (~63–65 °C) necessary for amplification. Fluorescent sensors positioned beneath each reaction chamber provide the real-time monitoring of the amplification progress, detecting signals emitted by the Eprimers upon target hybridization. Data from these sensors are processed by an internal control board, enabling the immediate interpretation of the results displayed directly on the device or transmitted wirelessly to external devices, such as smartphones or computers.

#### 5.2.3. Detection Time

In isothermal PCR, the reaction time is fast because the polymerase is synthesized continuously. Positive detection can be completed in 10 to 40 min, and a negative determination can be completed in about 40 min. Furthermore, there is a trade-off relationship between the time required to determine a negative test and the sensitivity, so if you are in a hurry to determine a negative test, it is okay to shorten the test time to about 25 to 30 min. By converting PCR reactions into POCT, various fields of application will expand, including outpatient treatment, intraoperative diagnoses, and remote diagnoses [13].

#### 5.2.4. Power Consumption and Portability

In isothermal PCR, as opposed to temperature-cycling PCR, only a heater is used and no Peltier device is used to raise or lower the temperature. Therefore, it does not consume power, making it possible to make the devices even smaller and more portable.

#### 5.2.5. Detection of Single-Base Mutations

The SmartAmp method is designed to allow for mutation-specific amplification, so DNA amplification itself can be used as a signal for mutation detection. Therefore, single nucleotide mutations can also be detected with the same device. Approximately 5% of resistance mutations occur in Tamiflu, the first-line drug for influenza virus infections. The main mutations are known to be one in NA1 (H274Y) and two in N2 (three if very low-frequency mutations are included) [13]. GenPad, which is based on the SmartAmp method, can identify resistant strains. In fact, if resistant strains are overlooked, severe symptoms such as influenza encephalopathy and influenza pneumonia can occur, so it is necessary to detect them early and change the therapeutic agent. Ideally, the entire process of detecting influenza and resistance mutations to Tamiflu would be completed within an hour, and we achieved this goal [28,29,30].

There is a known mutation in the virus resistant to the anti-influenza drug Tamiflu, one in NA1 (H274Y) and two in N2 (three if you include very low-frequency mutations). The amplification curves using NA1 wild strain (H274)-specific primers (blue) and mutant (H274Y)-specific primers (red) are shown. For the target RNA genome H274, amplification occurs only with the specific H274 primer, but not with the specific H274Y primer. For the target RNA genome H274Y, amplification occurs only with the H274Y-specific primer, but not with the H274-specific primer.

#### 5.2.6. Collection of Specimens

The pretreatment solution and process differ depending on the specimen to be collected, but basically, if the biological specimen can be collected using a cotton swab, the suspension in the pretreatment solution is placed into the cartridge [30,31]. For example, in the case of saliva, you can collect the sample by holding a cotton swab in your mouth, and then place the swab in the pretreatment solution (SSB solution). Basically, if a biological specimen is suspended in SSB, one can mail the specimen while still in the tube. SSB liquid completely inactivates viruses, so there is no risk of infection during transportation [13].

The new coronavirus culture strain is mixed into the GenPad SSB pretreatment solution, and immediately after treatment, low-molecular components are removed using a G25 span column. A series of dilutions is made from the excluded fraction, Velo cells cultured in a culture dish are infected, and the plaques are counted. Similar treatments were performed using PBS and a universal transport medium (UTM), and the inactivation of virus infectivity was measured by a plaque assay. SSB resulted in the complete inactivation of the virus [13].

#### 5.2.7. Closed System and Complete Automation

When the collected sample is injected, the cartridge cap is closed, and the start button is pressed. The entire process is fully automatic, from pretreatment to obtaining the results. Since it is a completely sealed system, the contamination of the amplicon is also prevented.

### 5.3. GenPad Clustering and Data Display

Since GenPad detects one sample with one cartridge, testing can be started immediately each time a sample arrives. When the knob is turned at the bottom of the device, the end of the connection protrudes from the bottom of the back (protruding connection plug) and can be connected to the (recessed) connection plug on the front of another device. Therefore, as shown in Figure 4, up to 8 PCs can be clustered in one row, and there can be up to 12 rows, that is, 8 PCs × 12 rows (96 PCs); therefore, any number of PCs from 1 to 96 PCs can be clustered at once. It can be controlled from one window. You can also connect your smartphone to a single GenPad device using BlueTooth and control the device. Therefore, a smartphone control system is optimal for personal (home) use. PC software (software version 0.6.11) (https://miraigenomics.com/device; accessed on 10 July 2025) can also be downloaded for free from the mobile app (GenPad).

Assuming social implementation, one sample may be measured by an individual, but there are also cases where each individual is expected to access it separately (random access). In such cases, the 96-well plate (8 wells × 12 wells) commonly used for testing is useless. This is because testing must wait from the time the first specimen arrives until the ninety-sixth specimen arrives. When multiple samples that arrive separately like this are submitted to the system, the analysis must begin immediately after each sample is collected. Clustered GenPad (GenPad Cluster) is perfect for the random-access purpose.

Eight GenPad devices can be connected in a row using the (protruding) connection plug on the back of the GenPad device and the (recessed) connection plug on the front of another device. The cartridges can be controlled by a PC in 12 rows of 8 cartridges (96 cartridges in total). In addition, if you want to control only one device, you can connect it to your smartphone using Bluetooth.

### 5.4. Comparison of GenPad and Other Testing Platforms

Table 2 shows the other current principles of rapidity, a high sensitivity, multiple test items, one-step (one-base) mutation detection, portability (small and lightweight), full automation (just flip a switch), and random access. This is a comparison with technology. GenPad is designed to meet all the criteria. We compared the characteristics of GenPad, conventional PCR machines, and immunochromatography (antigen testing) in terms of the portability, rapidity, sensitivity, simplicity (automation), mutation detection, and multi-target testing.

## 6. Potential Social Impact of GenPad-Like Systems

In 2021, Kanagawa Prefecture led a social demonstration study assuming the social implementation of GenPad. This was an attempt to test its usability and convenience by actually using it as an on-site test at private clinics, elderly care facilities, sports facilities (sports organizations), workplaces (inside companies), etc. Although various opinions were received, it was proven that it is extremely useful in situations where society is actually in trouble.

Although the initial pilot implementation of GenPad was conducted in Kanagawa Prefecture, similar systems have the potential to play a critical role in public health management worldwide. Applications of these technologies include disease outbreak control in remote or low-income areas, border health surveillance, sports and event management, and routine disease diagnoses in both developed and developing countries.

For example, in the workplace, it has been used to determine whether staff who are infected with the new coronavirus are recuperating at home or are on standby, even if they are asymptomatic, and can return to work. The timing at which germs cease to be excreted differs depending on the individual, but by using GenPad, it is possible to measure in a timely fashion the time when a negative change occurs, that is, when it is safe to return to work, using PCR-level sensitivity [37,38,39].

In addition, in a case where a new coronavirus infection occurred at a nursing home, 13 of the staff members were close contacts, and antigen tests were conducted immediately after the close contact and before much time had passed. Since all thirteen people tested negative, we considered the continuation of work, but when the same samples were measured using GenPad at the same time, nine people tested positive. This is an example in which cluster occurrence could be avoided [13].

We conducted the social implementation of GenPad on a trial and research basis. It was used to identify close contacts at elderly facilities, medical institutions (clinics), sports fields (soccer, baseball, etc.), companies, etc., to confirm negative results before events, and to determine the criteria for returning infected individuals to work. The purpose of its use and the comments of users are listed.

## 7. GenPad Is an Essential Element in the Construction of a Smart Medical City

### 7.1. Smart Medical City

Following the incident on the “Diamond Princess”, we need to consider urban systems that are prepared for future outbreaks of emerging infectious diseases. Furthermore, not only for emerging infectious diseases such as these, but also for other diseases, it is necessary to build a system that allows individuals to measure their health status in real time and communicate with medical institutions at all times. We call cities equipped with such systems “smart medical cities.” Data measured in hospitals, stations, emergency squads, homes, public places, airports, airplanes, and ships can always be communicated to a central medical institution, and in the event of an emergency, a doctor’s consultation can be received remotely. A system that can handle a wide variety of medical needs, not just emerging outpatient infectious diseases, is necessary [13].

### 7.2. Telemedicine and GenPad

In recent years, telemedicine has become widespread in daily life in urban areas. The social contribution that telemedicine can make is to prescribe drugs that are regularly taken daily by busy workers for lifestyle-related diseases, etc., and to reduce the congestion of medical institutions and the time required for patients to see a doctor. Telemedicine is becoming heavier and heavier. Furthermore, telemedicine has important implications for medical care on remote islands and depopulated areas without doctors. In particular, Japan, an island country with 432 remote islands inhabited, needs a system that can accurately measure the health status of individuals numerically, even for telemedicine targeting remote islands. In particular, these measurement systems need to be closely linked to the communication network infrastructure.

### 7.3. Potential Extension to Cancer Mutation Testing

In addition to infectious disease testing, the GenPad system has potential applications in oncology mutation testing. Prior studies have demonstrated that SmartAmp and Eprimer-based systems can detect somatic mutations, including KRAS mutations in colorectal cancer [40,41] and MDM2 SNPs linked to cancer risk [42]. These findings suggest that the GenPad platform could be customized to detect clinically relevant oncogenic mutations in outpatient or intraoperative settings, providing rapid molecular diagnostics at the point of care.

## 8. Conclusions

In summary, the GenPad system addresses multiple unmet needs in the current diagnostic practice by combining rapid, highly sensitive nucleic acid detection with single-base mutation identification, full automation, and portability. The system’s implementation during the course of the pandemic caused by the severe acute respiratory syndrome (SARS-CoV-2) virus, more commonly known as the coronavirus pandemic, revealed deficiencies in the existing healthcare infrastructure and demonstrated the potential of such systems for broader medical applications in addition to those related to infectious diseases. It is evident that GenPad, due to its versatility and scalability, represents a next-generation POCT platform that has the potential to make a substantial impact on both routine healthcare and emergency responses.

## Figures and Tables

**Figure 1 diagnostics-15-02020-f001:**
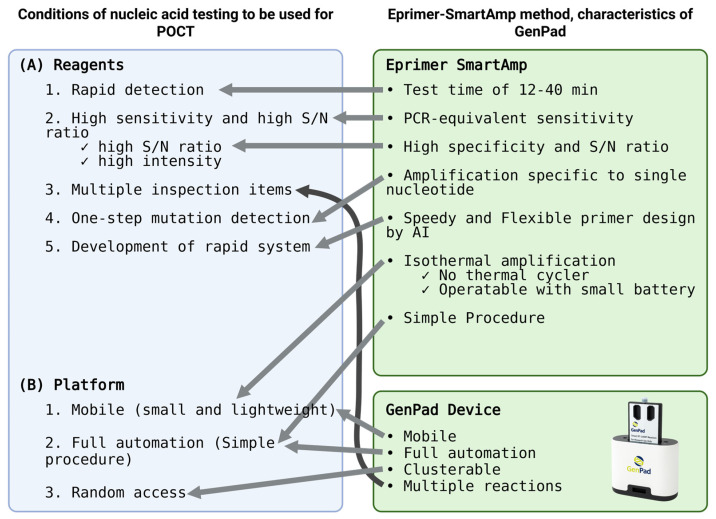
Matching of POCT needs in response to Japanese and global COVID-19 in GenPad system.

**Figure 2 diagnostics-15-02020-f002:**
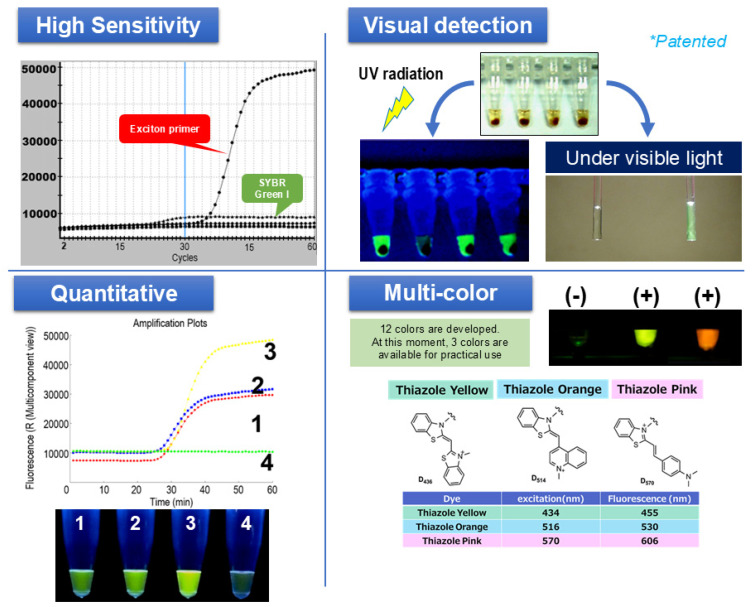
Principle and features of SmartAmp as a sensor for nucleic acids in the POCT system. **Upper left**: Demonstration of the high sensitivity of exciton primer (Eprimer^®^) technology compared to SYBR Green I, as indicated by steeper amplification curves. **Upper right**: Visual detection of positive (+) and negative (−) reactions under UV and visible light. **Bottom left**: Example of quantitative detection, where the fluorescence intensity directly correlates with the amplified product amount. **Bottom right**: Multi-color detection, illustrating three distinct fluorescent dyes (thiazole yellow, thiazole orange, and thiazole pink) enabling a multiplex analysis (Japanese Patent Application JP-2017-0107549).

**Figure 3 diagnostics-15-02020-f003:**
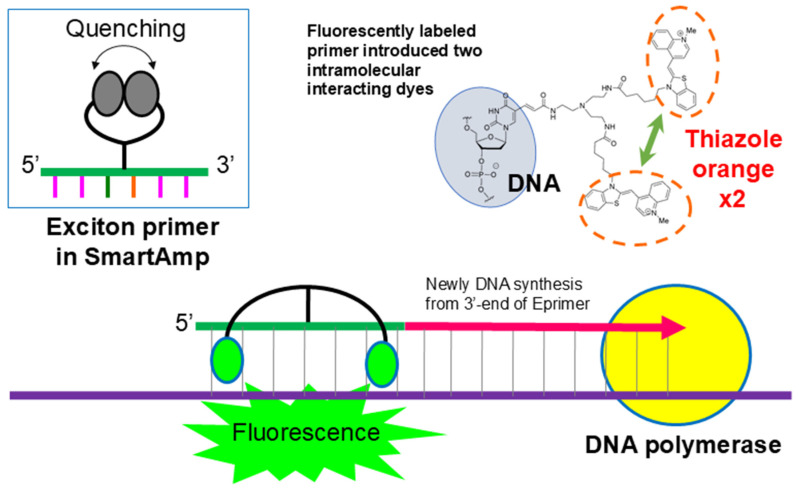
Principle of exciton primer (Eprimer^®^) technology utilized in GenPad POCT system. Exciton primers contain two intramolecularly interacting fluorescent dyes (e.g., thiazole orange). In the absence of target DNA, the dyes remain in close proximity, quenching fluorescence. Upon the specific hybridization of the primer to the complementary DNA sequence, the dyes separate, disrupting the quenching interaction and resulting in strong fluorescence emission. This mechanism enables the real-time, highly sensitive detection of amplified nucleic acids in SmartAmp-based diagnostic tests.

**Figure 4 diagnostics-15-02020-f004:**
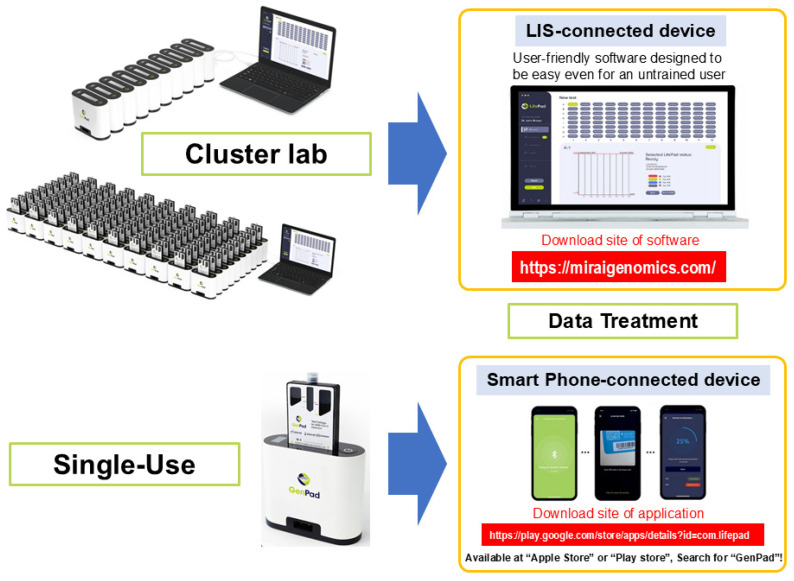
GenPad clustering and data display.

**Table 1 diagnostics-15-02020-t001:** Potential social impact of further integration of GenPad-like POCT systems.

Purpose/Action	Description of Application	Expected Outcome/Benefit	Application Field
Determination of positive/negative	Positive cases that were not detectable by on-site antigen testing were successfully identified. Compared to the usual PCR testing, it provides results much earlier; since conventional PCR tests are outsourced to a laboratory, it takes a while to receive results.	Preventing the spread of infection	Elderly facilities Medical institutes
	Prepared for the occurrence of fever cases in sports competitions.		Sports
	Rapid testing was successfully performed for employees who had close contact with positive cases.		Workplaces
Confirmation of negative	Causes other than COVID-19 were successfully identified, allowing for the provision of an appropriate treatment based on high-sensitivity COVID-19-negative confirmation.	Applying an appropriate treatment	Medical institutes
	Negative results were confirmed even in cases where infected individuals or close contacts were present in the vicinity.	Obtaining peace of mind: Maintaining daily life and social activities as much as possible	Users of elderly care facilities Workplaces
	It was possible to verify a negative transition prior to reintegration into society following illness and recovery.	Preventing the spread of infection	Workplaces
Testing of all participants	A testing procedure was conducted for all the participants of a sporting event.	Preventing the spread of infection	Sports

**Table 2 diagnostics-15-02020-t002:** Comparison of generalized detection approaches in GenPad and other testing platforms [32,33,34,35,36]. *—A direct conversion in copies/mL was not possible, since the ratio of “infectious units–genomic copies” is strain-, cell line-, and technique-dependent.

	Method	Time to Result (Min)	LOD (Copies or rxn/mL)
GenPad	Isothermal (SmartAmp)	10–40 pos/≤40 neg	10–50 rxn/ml
Xpert^®^ Xpress SARS-CoV-2	Microfluidic RT-PCR	30–60	≈100 copies/mL
Fluxergy Analyzer COVID-19	Microfluidic direct RT-PCR	≈65	0.3 TCID50/mL *
QIAstat-Dx Respiratory SARS-CoV-2 Panel	Multiplex microfluidic RT-PCR	60	500 copies/mL
Veros™ SARS-CoV-2 (Visby)	Isothermal (RT-LAMP)	30	100 copies/swab

## Data Availability

Not applicable.

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
