# Peer review of "GenPad: A Highly Efficient Roadmap for the Development of a New Rapid, Highly Sensitive, and Portable Point-of-Care Testing System for Nucleic Acid Diagnostics in Japan"

_diagnostics, 2025, doi:10.3390/diagnostics15162020_

Round 1
Reviewer 1 Report
Comments and Suggestions for Authors
The manuscript of diagnostics-3441888 entitled "GenPad: An Highly Efficient Roadmap for Development of a 2 New Rapid, High-Sensitive, and Portable POCT (Point of Care 3 Test) System for Nuclei Acid Diagnostics in Japan" . Author explores the importance of rapid detection of emerging infectious diseases, which is both timely and crucial. In the context of the current global crisis involving the novel coronavirus. But there are some details need to add in the manuscript.
1. Author have provided a detailed description of the Eprimer-SmatAmp technology and the GenPad system, which is crucial for readers to understand your research. However, some technical details may need to be further clarified or simplified to ensure that a broad readership can comprehend them. Please consider providing more background information, such as how can they amplified DNA? and what is The Turn Back primer?
2. The figure captions should be more detailed, as in Figure 2 and Figure 3, to help readers better understand the content of the images.
Author Response
Response to reviewers’ comments on the manuscript by Gusev O. entitled “GenPad: An Highly Efficient Roadmap for Development of a New Rapid, High-Sensitive, and Portable POCT (Point of Care Test) System for Nuclei Acid Diagnostics in Japan”
General Comment:
The manuscript of diagnostics-3441888 entitled "GenPad: An Highly Efficient Roadmap for Development of a 2 New Rapid, High-Sensitive, and Portable POCT (Point of Care 3 Test) System for Nuclei Acid Diagnostics in Japan" . Author explores the importance of rapid detection of emerging infectious diseases, which is both timely and crucial. In the context of the current global crisis involving the novel coronavirus. But there are some details need to add in the manuscript.
Response:
We would like to thank the reviewers for their thorough review of our manuscript. We are pleased with the generally positive comments. To address the reviewers’ concerns and excellent suggestions, we have revised the text based on the reviewers’ comments. We think that these changes have substantially strengthened the manuscript. We respond in detail to all the reviewers’ concerns below, and the changes in the manuscript are indicated by the using Track Changes mode
Comment 1:
Author have provided a detailed description of the Eprimer-SmatAmp technology and the GenPad system, which is crucial for readers to understand your research. However, some technical details may need to be further clarified or simplified to ensure that a broad readership can comprehend them. Please consider providing more background information, such as how can they amplified DNA? and what is The Turn Back primer?
Response:
We appreciate the reviewer’s insightful comment regarding the need for clearer explanation of the DNA amplification mechanism and the role of the Turn-Back primer. In response, we have added a new explanatory subsection to the manuscript (Section 3.1.1, lines 160-175). This section provides a concise and accessible description of the SmartAmp amplification process, including:
The principle of isothermal DNA amplification using a strand-displacing polymerase, without thermal cycling. The function of the Turn-Back primer (TB primer), which creates a self-folding loop that triggers rapid, continuous amplification under isothermal conditions. The mechanism also explains how single-base mismatches at the 3′-end prevent loop formation, ensuring single-nucleotide specificity. The roles of the Folding primer, Boost primer, and Exciton-labeled primer (Eprimer®) in facilitating and detecting amplification. This clarification is supported by key references, including Mitani et al. (Nat Methods, 2007, doi:10.1038/nmeth1007), Lezhava et al. (Hum Mutat, 2010, doi:10.1002/humu.21177), and Delobel et al. (BMC Infect Dis, 2022, doi:10.1186/s12879-022-07458-4), as well as recent clinical implementations in COVID-19 testing (Nagasawa et al., 2022; Asai et al., 2022).
Comment 2:
The figure captions should be more detailed, as in Figure 2 and Figure 3, to help readers better understand the content of the images
Response:
Thank you for this constructive comment. We have revised and expanded the captions for both Figure 2 and Figure 3 to ensure that each figure can be understood independently of the main text.
For Figure 2, we now provide a clearer explanation of the four panels, highlighting key features of the SmartAmp system: high sensitivity of Exciton Primer detection compared to SYBR Green I, the visual distinction between positive and negative samples, quantitative fluorescence read-out, and multi-color detection using different fluorophores (Thiazole Yellow, Orange, and Pink).
For Figure 3, we added a detailed description of the Exciton Primer (Eprimer®) mechanism. The figure illustrates how fluorescence is quenched when the dyes are in close proximity (non-hybridized state), and how hybridization with the target DNA separates the dyes, resulting in strong fluorescence emission. This mechanism enables real-time, highly specific signal generation directly coupled to nucleic acid amplification.
We believe these modifications address the reviewer’s concern and make the figures fully accessible to a broad readership.
Reviewer 2 Report
Comments and Suggestions for Authors
Reviewer Comments:
The author reported ‘GenPad: a roadmap for development of a new rapid, high-sensitive, and portable POCT (Point of Care Test) system for nuclei acid diagnostics in Japan’.
This study aimed to feature the rapid, high-sensitive, and portable POCT (Point of Care Test) system called GenPad for nuclei acid diagnostics. The manuscript actually highlighted the isothermal amplification of nucleic acid adopting full features (i.e., mechanism of assay, time of detection, efficiency, etc.) of Eprimer-SmatAmp technology, and connected the smartphone-integrated telemedicine for rapid management/surveillance of viral infectious diseases.
This manuscript features GenPad, its nucleic acid testing system, history of applications. Many of the features are already patented. So, it may be the company product. Just featuring its application seems not appealing for an article. It is neither a review nor a perspective, even not a case report. In the category of opinion, the structures of the manuscript should maintain scientific coherence regarding the presentation of introduction, methodology, recent development in similar instruments, advancement in the featured instruments, experimental/trialed/tested report analysis, comparison, shortcomings (if any). However, this manuscript has been organized as story-telling. The author should take decision to revise the contents to be published.
Counting all together, the manuscript is not suitable for publishing in diagnostics in the current format.
Major comments
1. In line 15 of the abstract, the author used the expression “we” which is not logical as this manuscript is single-authored i.e., written by an individual. The whole manuscript has many such expressions (“we”) that should be corrected or revised in different formats.
2. At least, a small introduction should be written highlighting the scope of this opinion or GenPad.
3. In case of featuring NAT device, the device and its could be exploded for better understanding.
4. In abstract, the author mentioned about mutation testing targeting cancer diagnosis. The body of text slightly touched it but data analysis or case reports are not well-presented.
5. Figure 2 presents patented content. It should be properly cited in the figure caption with copyright permission policy.
6. Are the data, photographs of devices, illustrations in Figure 2, 3 and 4 original? If yes, that’s fine. If not, citations are needed.
7. The repetition of texts should be avoided. For example, POC-NAT criteria has been mentioned in lines 46-49; which are also repeated in slightly modified form in lines 173-175 or somewhere else.
8. Over all, data presentation and discussion need to be more systematic.
9. The author should put evidence of mass tests in graphs, charts.
10. For nucleic acid tesing within 10-60 min, Several RT-PCR-based commercial kits/devices/panels (such as Star Array SARS-CoV- 2 Nucleic Acid Detection Kit 1.0, VitaPCR SARS-CoV-2 Assay, Accula SARS-CoV-2 test [NALFA], 2019-nCoV Nucleic Acid Diagnostic Kit, Portable 2019-nCoV diagnostic device [Microfluidic bioassay chip], Xpert® Xpress SARS-CoV-2 test, Truenat™ Beta CoV [Microluidic/Micro chip], Fluxergy Analyzer system, [Microfluidic], LexaGene’s automated LX Analyzer for SARS-CoV-2 [Microfluidic], QIAstat-Dx Respiratory SARS-CoV-2 Panel [Microfluidic] and others) are already reported. Similarly, isothermal platforms such as Abobott ID Now Covid 19 Test [Microfluidic], Veros™ SARS-CoV-2 test [Microfluidic], RapiPrep® COVID-19 test and others are reported for tests below 1 hour.
GenPad should be compared with them in terms of sensitivity, specificity, detection time, assay, cost and relevant other POC criteria.
11. Many public and private approaches are reported for smartphone-integrated test analysis and surveillance (Can be found in review articles). What is the originality of the GenPad’s smartphone integration?
12. The conceptual theme is workable throughout the globe. So it is not the matter of a Japanese city. The writeup should highlight the global perspective.
Comments on the Quality of English LanguageEnglish can be improved.
Author Response
Response to reviewers’ comments on the manuscript by Gusev O. entitled “GenPad: An Highly Efficient Roadmap for Development of a New Rapid, High-Sensitive, and Portable POCT (Point of Care Test) System for Nuclei Acid Diagnostics in Japan”
General Comment:
The author reported ‘GenPad: a roadmap for development of a new rapid, high-sensitive, and portable POCT (Point of Care Test) system for nuclei acid diagnostics in Japan’.
This study aimed to feature the rapid, high-sensitive, and portable POCT (Point of Care Test) system called GenPad for nuclei acid diagnostics. The manuscript actually highlighted the isothermal amplification of nucleic acid adopting full features (i.e., mechanism of assay, time of detection, efficiency, etc.) of Eprimer-SmatAmp technology, and connected the smartphone-integrated telemedicine for rapid management/surveillance of viral infectious diseases.
This manuscript features GenPad, its nucleic acid testing system, history of applications. Many of the features are already patented. So, it may be the company product. Just featuring its application seems not appealing for an article. It is neither a review nor a perspective, even not a case report. In the category of opinion, the structures of the manuscript should maintain scientific coherence regarding the presentation of introduction, methodology, recent development in similar instruments, advancement in the featured instruments, experimental/trialed/tested report analysis, comparison, shortcomings (if any). However, this manuscript has been organized as story-telling. The author should take decision to revise the contents to be published.
Counting all together, the manuscript is not suitable for publishing in diagnostics in the current format.
Response:
We would like to thank the reviewers for their thorough review of our manuscript. We acknowledge that the manuscript requires additional improvements to enhance its clarity and overall quality. To address the reviewers’ concerns and excellent suggestions, we have revised the text based on the reviewers’ comments. We think that these changes have substantially strengthened the manuscript. We respond in detail to all the reviewers’ concerns below, and the changes in the manuscript are indicated by the using Track Changes mode
Comment 1:
In line 15 of the abstract, the author used the expression “we” which is not logical as this manuscript is single-authored i.e., written by an individual. The whole manuscript has many such expressions (“we”) that should be corrected or revised in different formats.
Response:
We have reviewed every first-person reference and applied a consistent rule: Single-author statements that express the viewpoint or actions taken solely for this Opinion piece have been rewritten in the singular “I.” or impersonal. Example (Section 1, paragraph 4):
Original: “In this article, we would like to look back and summarize what a small team has learnt …”
Revised: “This article summarizes the lessons learned from facing the coronavirus pandemic that started with Diamond Princess”
Descriptions of historical or experimental work carried out collectively—for instance, the RIKEN group’s development of SmartAmp/GenPad—retain the plural “we/our team,” because those activities involved several investigators. Example (same section, earlier sentence):
“After receiving a strong request to make a new coronavirus test kit, we immediately began developing new coronavirus test reagents.” All other occurrences have been adjusted accordingly; the tracked-changes file shows each edit. We trust this resolves the stylistic issue.
Comment 2:
At least, a small introduction should be written highlighting the scope of this opinion or GenPad.
Response:
Thank you for this valuable suggestion. In the revised manuscript, we have added a new Introduction section at the beginning of the text (lines 24-43).
This introduction provides:
- The context and motivation for developing rapid, portable nucleic acid diagnostics, emphasizing the challenges faced during the COVID-19 pandemic and the “Diamond Princess” incident.
- A clear statement of the scope and aim of this Opinion article: to discuss the GenPad system based on SmartAmp-Eprimer technology, outline its technological features, compare it with existing POCT platforms, and consider its future applications in telemedicine and public health.
- An explanation of the article’s structure, guiding the reader through the technological background, practical implementation, comparison, and outlook.
We believe this addition improves the clarity and positioning of the manuscript, as requested.
Comment 3:
In case of featuring NAT device, the device and its could be exploded for better understanding.
Response:
Thank you for this important suggestion. While we did not include an additional exploded diagram figure, we have expanded the manuscript to provide a clear and detailed textual description of the GenPad device’s internal components and workflow.
Specifically, we have added a new paragraph in Section 4.2.2 (255-267), following the description of the cartridge. This new text explains:
- The structure of the disposable cartridge, containing four reaction chambers preloaded with reagents;
- The integrated heating module for maintaining isothermal amplification conditions;
- The placement of fluorescent sensors for real-time signal detection;
- The data processing unit for immediate interpretation and transmission of results to external devices such as smartphones or computers.
We believe this addition satisfies the reviewer’s request for a clearer understanding of the GenPad platform’s design and function, even in the absence of an exploded schematic figure.
Comment 4:
In abstract, the author mentioned about mutation testing targeting cancer diagnosis. The body of text slightly touched it but data analysis or case reports are not well-presented
Response:
We thank the reviewer for raising this point. As this is an Opinion article, the primary focus of the manuscript is on the technological principles and potential applications of the GenPad system, rather than presenting new oncology case data.
To clarify the relevance of cancer mutation testing, we have added a dedicated subsection in the revised manuscript in Section 6.3 (lines 415-422). This section explicitly discusses how the GenPad system could be adapted for somatic mutation detection in oncology, based on prior published applications of SmartAmp technology.
Specifically, we cite:
- Tanaka et al., Oncol Rep 2014, demonstrating KRAS mutation detection using SmartAmp-Eprimer technology (DOI: 10.3892/or.2014.3487).
- Enokida et al., PLoS ONE 2013, reporting rapid detection of the MDM2 c.309T>G cancer-associated SNP using the duplex SmartAmp method (DOI: 10.1371/journal.pone.0060151).
We emphasize that these studies validate the core chemistry for cancer mutation applications, supporting the potential future use of the GenPad platform in oncology. While this manuscript does not present new experimental data in that field, we believe that including this discussion aligns with the scope of a perspective piece.
Comment 5:
Figure 2 presents patented content. It should be properly cited in the figure caption with copyright permission policy.
Response:
Figure 2 illustrates the conceptual mechanism of the SmartAmp method and Eprimer technology. These technologies are indeed patented, and the figure was specifically created for this manuscript by the authors. We have revised the caption of Figure 2 to include the relevant patent reference. The figure now explicitly acknowledges that the depicted method is based on patented technology (Japanese Patent Application JP‑2017‑0107549). Since the illustration was created by the authors specifically for this manuscript, no third-party copyrighted materials are used.
Comment 6:
Are the data, photographs of devices, illustrations in Figure 2, 3 and 4 original? If yes, that’s fine. If not, citations are needed
Response:
The illustrations in Figures 2, 3, and 4 are original and were specifically created for this manuscript. this is conceptual illustrations intended to explain the principles and design of the system.
Comment 7:
The repetition of texts should be avoided. For example, POC-NAT criteria has been mentioned in lines 46-49; which are also repeated in slightly modified form in lines 173-175 or somewhere else.
Response:
We would like to thank the reviewer for highlighting this redundancy. we confirmed that the POC-NAT criteria were mentioned in both the introduction and subsequent sections.
In response to this comment, a thorough review of the manuscript was conducted, resulting in the elimination of superfluous descriptions. The detailed explanation of the POC-NAT criteria has been retained in the introduction, in order to provide a context for the paper. In subsequent parts of the manuscript, sections pertinent to the subject matter have been organised in order to avoid unnecessary repetition, while maintaining logical consistency and clarity (220-224).
We have also checked the rest of the manuscript to minimize similar repetitions. We believe this resolves the concern.
Comment 8:
Over all, data presentation and discussion need to be more systematic
Response:
We are indebted to the reviewer for their invaluable feedback regarding the necessity for more systematic data presentation and discussion. In response, we have made numerous corrections, clarifications, and structural improvements to the manuscript. These alterations are meticulously markedin the revised version through the utilisation of the change tracking mode.
We hope that these changes will have significantly improved the clarity, coherence, and overall systematic presentation of the material.
Comment 9:
The author should put evidence of mass tests in graphs, charts.
Response:
It is acknowledged that the presentation of large-scale test data in graphical or tabular form would enhance the comprehensibility of the findings. However, at this stage, the current manuscript is intended as an Opinion article focusing on the system concept, the rationale for its design, and lessons learned from early deployment.
A separate original article is in preparation, providing detailed data from mass testing and field validation. As these data are planned to be published in a subsequent manuscript, they cannot be included at this time due to concerns regarding publication and data confidentiality.
Comment 10:
For nucleic acid tesing within 10-60 min, Several RT-PCR-based commercial kits/devices/panels (such as Star Array SARS-CoV- 2 Nucleic Acid Detection Kit 1.0, VitaPCR SARS-CoV-2 Assay, Accula SARS-CoV-2 test [NALFA], 2019-nCoV Nucleic Acid Diagnostic Kit, Portable 2019-nCoV diagnostic device [Microfluidic bioassay chip], Xpert® Xpress SARS-CoV-2 test, Truenat™ Beta CoV [Microluidic/Micro chip], Fluxergy Analyzer system, [Microfluidic], LexaGene’s automated LX Analyzer for SARS-CoV-2 [Microfluidic], QIAstat-Dx Respiratory SARS-CoV-2 Panel [Microfluidic] and others) are already reported. Similarly, isothermal platforms such as Abobott ID Now Covid 19 Test [Microfluidic], Veros™ SARS-CoV-2 test [Microfluidic], RapiPrep® COVID-19 test and others are reported for tests below 1 hour.
GenPad should be compared with them in terms of sensitivity, specificity, detection time, assay, cost and relevant other POC criteria.
Response:
Numerous RT-PCR and isothermal nucleic acid detection systems have been developed for the purpose of rapid pathogen detection, with many of these systems capable of operating within a time frame of 10-60 minutes. However, these systems are often focused on single aspects, such as speed or portability, and typically require trade-offs in other parameters, such as mutation detection capability or workflow flexibility.
The GenPad platform has been designed to integrate multiple features into a single system, including high sensitivity comparable to laboratory PCR (10-50 copies per reaction), fast detection (up to 12-15 minutes for positive results), one-step single base mutation detection, full automation from sample input to results output, and true portability. Furthermore, GenPad facilitates random access operation and scalable clustering, enabling immediate testing of incoming samples without the need for batching.
While a direct comparison of cost and performance with all available commercial platforms is beyond the scope of this paper, Table 1 conceptually summarises the differences between GenPad and other systems by illustrating the combination of features implemented in this platform. The present study focuses on the experience of system integration and social implementation, especially during the course of the pandemic of the Coronavirus (SARS-CoV-2) in Japan, which emphasised the practical need for such solutions.
Comment 11:
Many public and private approaches are reported for smartphone-integrated test analysis and surveillance (Can be found in review articles). What is the originality of the GenPad’s smartphone integration?
Response:
We thank the reviewer for this important question. In the GenPad system, smartphone integration serves as a control and data management tool. As described in the manuscript (Section 4.3, Figure 4), a smartphone can be connected to a single GenPad device via Bluetooth to initiate testing and to control the process in personal (home) use.
Furthermore, the system allows clustering of multiple GenPad devices (up to 96 units) through physical connection plugs, with centralized management via a PC interface. The smartphone control option is specifically implemented for single-unit operation, providing flexibility for individual use cases.
Comment 12:
The conceptual theme is workable throughout the globe. So, it is not the matter of a Japanese city. The writeup should highlight the global perspective.
Response:
We would like to thank the reviewer for this valuable comment. It is acknowledged that the notion of portable, automated, random-access POCT systems possesses considerable global relevance, transcending geographical boundaries and thus being applicable beyond a specific region.
In response to the aforementioned comment, the manuscript was revised to more clearly emphasise the global applicability of the GenPad platform (lines 416-420). Although the preliminary development and implementation occurred in Japan, the challenges addressed, such as the decentralised testing and real-time health monitoring, are of universal significance.
Reviewer 3 Report
Comments and Suggestions for Authors
The author described a Point-of-Care Test (POCT) system for emerging disease detection and control, but it is really poorly written. The figure feels like just screenshot from some ppt slides and there is even not a formal introduction. There are also not scientific discussions about the working pipelines or mechanisms. I found it really hard to read this paper and didn’t get the novelty or impact of this work. I think it should be directly rejected.
Comments on the Quality of English LanguageThe author described a Point-of-Care Test (POCT) system for emerging disease detection and control, but it is really poorly written. The figure feels like just screenshot from some ppt slides and there is even not a formal introduction. There are also not scientific discussions about the working pipelines or mechanisms. I found it really hard to read this paper and didn’t get the novelty or impact of this work. I think it should be directly rejected.
Author Response
Response to reviewers’ comments on the manuscript by Gusev O. entitled “GenPad: An Highly Efficient Roadmap for Development of a New Rapid, High-Sensitive, and Portable POCT (Point of Care Test) System for Nuclei Acid Diagnostics in Japan”
We would like to thank the reviewer for their thorough review of our manuscript. We have revised the text based on the reviewers’ comments. We think that these changes have substantially strengthened the manuscript. We respond in detail to reviewers’ concerns below.
General Comment:
The author described a Point-of-Care Test (POCT) system for emerging disease detection and control, but it is really poorly written. The figure feels like just screenshot from some ppt slides and there is even not a formal introduction. There are also not scientific discussions about the working pipelines or mechanisms. I found it really hard to read this paper and didn’t get the novelty or impact of this work. I think it should be directly rejected.
Response:
We sincerely thank the reviewer for their careful reading and constructive evaluation of our manuscript. We fully agree with the reviewer’s comment regarding the figures. In response, we have revised Figure 1 and reformatted Figures 5 and 6 into a Table 1 and 2 to enhance clarity and readability. In accordance with the reviewer’s observation, a distinct introduction section has been incorporated into the manuscript to provide clearer background and context (lines 24-43). In addition, revisions have been made directly within the manuscript (track changes mode) to strengthen the clarity and overall quality of the text.
We acknowledge that, as this manuscript is categorized as an “Opinion paper”, the primary aim is not to introduce a completely new methodology, but to highlight the lessons learned from real-world implementation of POCT systems during public health emergencies and to propose a conceptual framework for future healthcare preparedness. Nevertheless, we would like to clarify novel aspects and societal impact emphasized in this work: The GenPad platform enables one-step detection of single nucleotide mutations at the point of care, using the SmartAmp method combined with Eprimer technology. This allows for real-time identification of resistant viral strains (e.g., Tamiflu-resistant influenza variants), which is not currently achievable with standard POCT systems. GenPad supports random-access clustering mechanism, allowing immediate individual testing without batching – this is particularly relevant for outbreak management in real-life settings such as workplaces, sports venues, and elderly care facilities. The platform was actually deployed and tested in multiple societal contexts (elderly care, sports, workplaces) during the COVID-19 pandemic in Kanagawa Prefecture, providing practical insights beyond bench validation. This experience is rarely captured in academic literature and represents an important translational contribution.
To address the general comment of the reviewer, we also have revised the manuscript by invention of the conclusion section to more clearly highlight the lack of structure of the manuscript (lines 424-434).
Reviewer 4 Report
Comments and Suggestions for Authors
The article is an opinion-based, narrative, and descriptive piece discussing a rapid, portable, and sensitive diagnostic system called GenPad, which is built on the Eprimer-SmartAmp technology. This technique uses isothermal amplification of nucleic acids to detect single-point mutations. The article explores and reflects on the experiences and lessons learned during the COVID-19 pandemic.
It describes a relatively underused method that deserves broader dissemination.
While the article provides explanations about how the technique functions, it could benefit from additional figures and more detailed descriptions of the Eprimer-SmartAmp technology, as well as further clarification of what GenPad is and how it operates.
Including illustrative images, particularly of the system’s internal components, would significantly enhance the reader's understanding. Although the author references multiple sources, including such visuals and explanations would make the article more comprehensive.
Additionally, in Figure 2, the author notes that something is patented but does not clarify whether it is the Eprimer-SmartAmp technology itself or the detection system described in the figure. Providing clarity on what is covered by patents would be important for understanding the commercial potential of both the technology and its findings.
The manuscript lack detailed information about the sensitivity, specificity, and overall performance of the method. The article lacks tables, graphs, or comparative studies with established methods such as qPCR.
I recommend the publication of this article after these adjustments have been made.
Author Response
Response to reviewers’ comments on the manuscript by Gusev O. entitled “GenPad: An Highly Efficient Roadmap for Development of a New Rapid, High-Sensitive, and Portable POCT (Point of Care Test) System for Nuclei Acid Diagnostics in Japan”
We would like to thank the reviewer for their thorough review of our manuscript. We have revised the text based on the reviewers’ comments. We think that these changes have substantially strengthened the manuscript. We respond in detail to reviewers’ concerns below.
General Comment:
The article is an opinion-based, narrative, and descriptive piece discussing a rapid, portable, and sensitive diagnostic system called GenPad, which is built on the Eprimer-SmartAmp technology. This technique uses isothermal amplification of nucleic acids to detect single-point mutations. The article explores and reflects on the experiences and lessons learned during the COVID-19 pandemic.
It describes a relatively underused method that deserves broader dissemination.
I recommend the publication of this article after these adjustments have been made.
Response:
We are pleased with the generally positive comments. To address the reviewers’ concerns and excellent suggestions, we have revised the text on the reviewers’ comments
Comment 1:
While the article provides explanations about how the technique functions, it could benefit from additional figures and more detailed descriptions of the Eprimer-SmartAmp technology, as well as further clarification of what GenPad is and how it operates.
Including illustrative images, particularly of the system’s internal components, would significantly enhance the reader's understanding. Although the author references multiple sources, including such visuals and explanations would make the article more comprehensive.
Response:
We appreciate the reviewer’s insightful comment regarding the need for clearer explanation of the DNA amplification mechanism and the role of the Turn-Back primer. In response, we have added a new explanatory subsection to the manuscript (Section 3.1.1, lines 160-175). This section provides a concise and accessible description of the SmartAmp amplification process, including:
While we did not include an additional exploded diagram figure, we have expanded the manuscript to provide a clear and detailed textual description of the GenPad device’s internal components and workflow.
Additionally, we have added a new paragraph (Section 4.2.2 lines 259-271), following the description of the cartridge. This new text explains:
- The structure of the disposable cartridge, containing four reaction chambers preloaded with reagents;
- The integrated heating module for maintaining isothermal amplification conditions;
- The placement of fluorescent sensors for real-time signal detection;
- The data processing unit for immediate interpretation and transmission of results to external devices such as smartphones or computers.
We believe this addition satisfies the reviewer’s request for a clearer understanding of the Eprimer-SmartAmp technology and GenPad platform’s design and function, even in the absence of an exploded schematic figure.
Furthermore, the description of Figures 2 and 3 has been expanded and clarified in order to facilitate a more profound comprehension of the mechanisms and principles of the technology.
Comment 2:
Additionally, in Figure 2, the author notes that something is patented but does not clarify whether it is the Eprimer-SmartAmp technology itself or the detection system described in the figure. Providing clarity on what is covered by patents would be important for understanding the commercial potential of both the technology and its findings.
Figure 2 illustrates the conceptual mechanism of the SmartAmp method and Eprimer technology. These technologies are indeed patented, and the figure was specifically created for this manuscript by the authors. We have revised the caption of Figure 2 to include the relevant patent reference. The figure now explicitly acknowledges that the depicted method is based on patented technology (Japanese Patent Application JP‑2017‑0107549). Since the illustration was created by the authors specifically for this manuscript, no third-party copyrighted materials are used.
Response:
Figure 2 illustrates the conceptual mechanism of the SmartAmp method and Eprimer technology. These technologies are indeed patented, and the figure was specifically created for this manuscript by the authors. We have revised the caption of Figure 2 to include the relevant patent reference. The figure now explicitly acknowledges that the depicted method is based on patented technology (Japanese Patent Application JP‑2017‑0107549). Since the illustration was created by the authors specifically for this manuscript, no third-party copyrighted materials are used.
Comment 3:
The manuscript lack detailed information about the sensitivity, specificity, and overall performance of the method. The article lacks tables, graphs, or comparative studies with established methods such as qPCR.
Response:
Numerous RT-PCR and isothermal nucleic acid detection systems have been developed for the purpose of rapid pathogen detection, with many of these systems capable of operating within a time frame of 10-60 minutes. However, these systems are often focused on single aspects, such as speed or portability, and typically require trade-offs in other parameters, such as mutation detection capability or workflow flexibility.
The GenPad platform has been designed to integrate multiple features into a single system, including high sensitivity comparable to laboratory PCR (10-50 copies per reaction), fast detection (up to 12-15 minutes for positive results), one-step single base mutation detection, full automation from sample input to results output, and true portability. Furthermore, GenPad facilitates random access operation and scalable clustering, enabling immediate testing of incoming samples without the need for batching.
While a direct comparison of cost and performance with all available commercial platforms is beyond the scope of this paper, Table 1 conceptually summarises the differences between GenPad and other systems by illustrating the combination of features implemented in this platform. The present study focuses on the experience of system integration and social implementation, especially during the course of the pandemic of the Coronavirus (SARS-CoV-2) in Japan, which emphasised the practical need for such solutions.
Round 2
Reviewer 2 Report
Comments and Suggestions for Authors
Reviewer Comments on Revised Manuscript:
The author attempted to address many of the queries. Yet, there is scope to improvise. The author is requested to address the following major revisions.
Major comments
- Line 3: In the title, the word “Nuclei” should be “Nucleic”.
- The expressions “Point-of-Care Test (POCT)” should be similar throughout the manuscript. For example, in the title, it is expressed as ‘Point of Care Test (POCT)’. Moreover, in the title, it is recommended to use either abbreviated or elaborated form (not to use both forms). Also, “Test” or “Testing” expression may need attention after POC, as “Testing” is usually followed for a system.
- Line 13-15: These lines in abstract seem not grammatically correct.
- The introduction is very short and lacks citations. The author is advised to follow the journal instructions. If the journal allows this, it is fine.
- In Figure 2, the small tubes showing fluorescence signal seem experimental output. Have those previously been used in any manuscript, thesis, patents, or are they original? If yes, the figure caption should acknowledge with citations/references.
- In Table 1, the expression “we” remained, which could be changed. The author is requested to check for any others if present. Note: ‘we’ expression is very general, can be used to mention authors’ previous works. However, this time, your team is not present as authors. That is why the expression should go formally.
- The previous comment number 10 has not been properly addressed. A table featuring the GenPad and the current commercial and private RT-PCR or isothermal techniques should be compared in terms of sensitivity, specificity, detection time, assay, cost and relevant other POC criteria. Please, focus on numeric values of those tests or systems. The table should be specific, not general writing in the manuscript. It is critical to understand the value of GenPad over others.
Author Response
Response to reviewers’ comments on the manuscript by Gusev O. entitled “GenPad: An Highly Efficient Roadmap for Development of a New Rapid, High-Sensitive, and Portable Point-of-Care Testing System for Nucleic Acid Diagnostics in Japan”
General Comment
The author attempted to address many of the queries. Yet, there is scope to improvise. The author is requested to address the following major revisions.
Response:
We would like to thank the reviewer for their re-evaluation of manuscript. We have revised the text based on the reviewers’ comments. We sincerely appreciate your constructive feedback. Please find our detailed responses to each of comments below.
Comment 1
Line 3: In the title, the word “Nuclei” should be “Nucleic”.
Response:
We have corrected this phrase (line 4)
Comment 2
The expressions “Point-of-Care Test (POCT)” should be similar throughout the manuscript. For example, in the title, it is expressed as ‘Point of Care Test (POCT)’. Moreover, in the title, it is recommended to use either abbreviated or elaborated form (not to use both forms). Also, “Test” or “Testing” expression may need attention after POC, as “Testing” is usually followed for a system.
Response:
We would like to express our gratitude for the precise comment on the use of terminology and abbreviations in the text. The utilisation of the phrases 'point-of-care' and 'POCT' has been standardised through the text, and the title of the manuscript has been revised.
Comment 3
Line 13-15: These lines in abstract seem not grammatically correct.
Response:
We acknowledge the inaccuracy in the text (lines 13-15) and thank the reviewer for the valuable comment. This section of the abstract (lines 13-15) has been revised and all inaccuracies have been rectified.
Comment 4
The introduction is very short and lacks citations. The author is advised to follow the journal instructions. If the journal allows this, it is fine.
Response:
We acknowledge the reviewer’s comment and appreciate the important note regarding the brevity of the introduction and the limited citation of literature. In response, the introduction has been revised to include additional textual information and supporting references – lines 33-35 and 38-40.
Comment 5
In Figure 2, the small tubes showing fluorescence signal seem experimental output. Have those previously been used in any manuscript, thesis, patents, or are they original? If yes, the figure caption should acknowledge with citations/references.
Response:
SmartAmp method and Eprimer technology fundamental principles representation shown at Figure 2 and these methods are covered by intellectual property protections, and the figure itself was newly created by the authors specifically for inclusion in this manuscript. In order to clarify this, the figure caption has been updated to reference the relevant patent (Japanese Patent Application JP‑2017‑0107549), explicitly noting that the illustrated methodology is derived from patented technology. No third-party materials or previously published figures have been used in the preparation of this illustration.
Comment 6
In Table 1, the expression “we” remained, which could be changed. The author is requested to check for any others if present. Note: ‘we’ expression is very general, can be used to mention authors’ previous works. However, this time, your team is not present as authors. That is why the expression should go formally.
Response:
We thank the reviewer for this valuable observation. The text in Table 1 (column – Description of Application) has been revised accordingly, and all expressions have been formulated in a more formal and impersonal manner
Comment 7:
The previous comment number 10 has not been properly addressed. A table featuring the GenPad and the current commercial and private RT-PCR or isothermal techniques should be compared in terms of sensitivity, specificity, detection time, assay, cost and relevant other POC criteria. Please, focus on numeric values of those tests or systems. The table should be specific, not general writing in the manuscript. It is critical to understand the value of GenPad over others.
Response:
We thank the reviewer for reiterating the importance of a quantitative, side-by-side comparison. In response, Table 1 has been completely revised: GenPad is now compared against representative commercial RT-PCR and isothermal POCT systems for SARS-CoV-2, using published numeric data for method, limit of detection and time-to-result.The criterion for including systems for comparison with genpad was the availability of data for comparison.
Regarding pertest cost, precise values are not reported by most manufacturers and vary widely with market, volume, and regional procurement contracts. Because of that we didn't include test cost in our table 1.
Reviewer 3 Report
Comments and Suggestions for Authors
It can be accepted now
Author Response
Response to reviewers’ comments on the manuscript by Gusev O. entitled “GenPad: An Highly Efficient Roadmap for Development of a New Rapid, High-Sensitive, and Portable Point-of-Care Testing System for Nucleic Acid Diagnostics in Japan”
General Comment
It can be accepted now
Response:
We would like to thank the reviewer renewed assessment of our manuscript. We greatly appreciate your constructive input during the review process; your comments have helped us refine and strengthen the work, making the final version more persuasive and rigorous.
Round 3
Reviewer 2 Report
Comments and Suggestions for Authors
The author attempted revising the manuscript.
Adding GenPad's features alongside commercial techniques in Table 2 could be appealing.